# Cellular Heterogeneity–Adjusted cLonal Methylation (CHALM) improves prediction of gene expression

Jianfeng Xu[1,2,6], Jiejun Shi[1,6], Xiaodong Cui[2], Ya Cui [1], Jingyi Jessica Li [3], Ajay Goel [4], Xi Chen [2], Jean-Pierre Issa [5✉], Jianzhong Su[2✉] & Wei Li [1✉]

Promoter DNA methylation is a well-established mechanism of transcription repression, though its global correlation with gene expression is weak. This weak correlation can be attributed to the failure of current methylation quantification methods to consider the heterogeneity among sequenced bulk cells. Here, we introduce Cell Heterogeneity–Adjusted cLonal Methylation (CHALM) as a methylation quantification method. CHALM improves understanding of the functional consequences of DNA methylation, including its correlations with gene expression and H3K4me3. When applied to different methylation datasets, the CHALM method enables detection of differentially methylated genes that exhibit distinct biological functions supporting underlying mechanisms.

[1] Division of Computational Biomedicine, Department of Biological Chemistry, School of Medicine, University of California, Irvine, CA 92697, USA. [2] Department of Molecular and Cellular Biology, Baylor College of Medicine, Houston, TX 77030, USA. [3] Department of Statistics, University of California, Los Angeles, CA 90095, USA. [4] Department of Molecular Diagnostics and Experimental Therapeutics, Beckman Research Institute of City of Hope, Duarte, CA 91010, USA. [5] The Coriell Institute for Medical Research, Camden, NJ 08103, USA. [6] These authors contributed equally: Jianfeng Xu, Jiejun Shi. ✉email: jpissa@coriell.org; jianzhongsu82@gmail.com; wei.li@uci.edu

D NA methylation is an essential epigenetic modification, and its role in transcription repression has been widely studied for decades. Jones et al[1]. demonstrated that methylated CpGs (mCpGs) in the promoter region are recognized by methyl-CpG-binding domain (MBD) proteins, which subsequently recruit histone deacetylase complexes to repress downstream gene expression[1]. Paradoxically, almost all studies conducted using genome-wide methylation profiling technologies such as whole-genome bisulfite sequencing (WGBS) have demonstrated a poor global correlation between promoter methylation and gene expression[2–5]. For example, Booth et al[2]. found only slightly negative correlations between transcription and both 5mC and 5hmC levels in promoter CpG islands (CGIs), whose role in transcription regulation has been well-established[6]. Efforts to address this paradox have shown that complex methylation patterns of regions much longer than promoters[7,8] (e.g., a 10-kb window surrounding the transcription start site) better explain gene expression. However, why promoter methylation alone is only weakly correlated with gene expression has not been directly addressed.

Here, we show that the poor correlation between promoter methylation and gene expression is due in part to the overly simplistic nature of the traditional DNA methylation quantification method (i.e., it determines just the mean methylation level of every CpG within a promoter)[9]. A key disadvantage of this traditional method is that it fails to account for heterogeneity among sequenced bulk cells but treats CpGs within or across cells as if they are identical (Supplementary Fig. 1a). For example, 20% of the cells in population A (Fig. 1a) are fully methylated in a promoter region, whereas the rest of the cells are fully unmethylated. In cell population B (Fig. 1b), there is one mCpG site per promoter in every cell. The traditional quantification method would indicate that the methylation level of this promoter is the same in both populations. Nevertheless, as previous studies demonstrated that a single mCpG is sufficient for recruiting MBD proteins[10,11] for gene repression, we hypothesized that this promoter would be repressed in 20% and 100% of cells in these two populations, respectively. Apparently, the traditional method fails to capture the potential expression difference. To avoid this pitfall, we developed a methylation quantification method: Cell Heterogeneity–Adjusted cLonal Methylation (CHALM), which leverages the fact that each bisulfite sequencing read likely represents a single cell within the sequenced bulk cells. Clonal methylation here refers to the binary methylation status (methylated or unmethylated) of a genomic locus in a single cell (represented by a read in bisulfite sequencing data).

Instead of calculating the mean methylation level of all CpG sites, CHALM quantifies the promoter methylation as the ratio of methylated reads (with ≥1 mCpG) to total reads mapped to a given promoter region. According to CHALM, the promoter methylation levels of these two cell populations would be 0.2 and 1, which might better explain the transcription activity. As expected, on promoter CGIs, CHALM-determined methylation levels fit a bimodal distribution (Supplementary Fig. 1b) and are usually higher than traditionally determined methylation levels (Fig. 1c). We show that the CHALM method improves the prediction of transcription activities by examining its correlation with gene expression and H3K4me3 level. Further comparisons between CHALM and the traditional method indicate that our method is capable of identifying more accurate differentially methylated genes that exhibit distinct biological functions supporting underlying mechanisms.

## Results

**CHALM better predicts the gene expression and H3K4me3 level in promoter CGIs.** For methods comparison, we mainly focus on promoter CGIs (Supplementary Data 1), which have been extensively studied for the relationship between DNA methylation and gene expression. We first assessed the power of the CHALM method in terms of predicting gene expression on a genome-wide scale using a CD3 primary cell dataset. Although the methylation levels calculated by both CHALM and traditional methods were anti-correlated with gene expression (Fig. 2a and Supplementary Figs. 2a, 3a, 3c, and 4), the CHALM-determined methylation levels exhibited a more linear and monotonic relationship with gene expression. As expected, lowly methylated promoter CGIs exhibited a very weak correlation between traditional methylation and gene expression[12–14] (Fig. 2b and Supplementary Figs. 2b and 4). Surprisingly, we observed a much stronger correlation between gene expression and CHALM-determined methylation (Fig. 2b and Supplementary Figs. 2c and 4). In addition, although we primarily focused on promoter CGIs (Methods), CHALM also outperformed the traditional methods in several other widely studied genomic regions (Supplementary Figs. 5–7).

DNA methylation is also known to be mutually exclusive with H3K4me3, which is strongly associated with gene expression. Unmethylated H3K4 is capable of releasing the auto-inhibition of DNMT3A by disrupting the interaction between the ATRX-DNMT3-DNMT3L and catalytic domains, thereby inducing de novo methylation[15,16]. We therefore examined the relationship between DNA methylation and H3K4me3 level in promoter

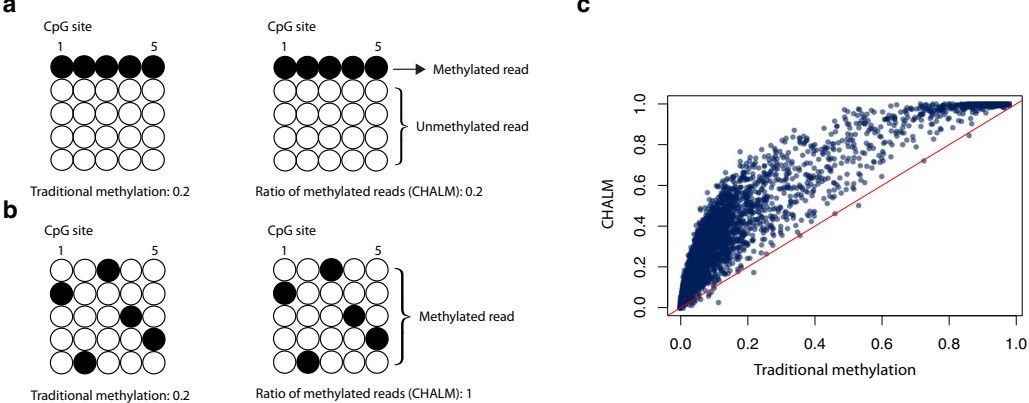

**Fig. 1 CHALM quantifies cell heterogeneity–adjusted DNA methylation level. a**, **b** show two different methylation patterns of a promoter region that cannot be distinguished by the traditional method. **c** Scatter plot shows a comparison of the methylation level calculated by the traditional and CHALM methods for the promoter CGIs of CD3 primary cells.

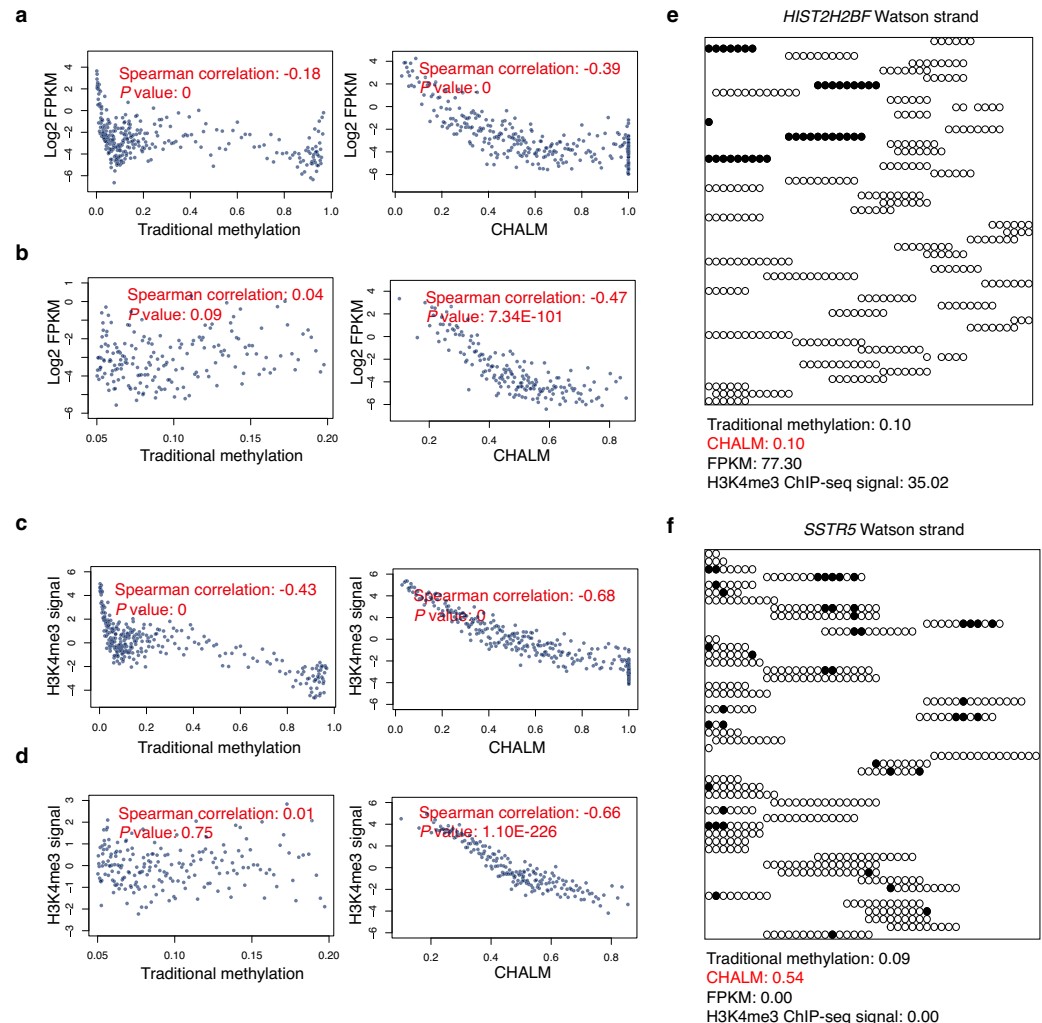

**Fig. 2 The CHALM method better predicts gene expression. a** Scatter plots show the correlation between gene expression and methylation level calculated using both methods. Balanced promoter CGIs (Methods section) of CD3 primary cells are used. Each data point represents the average value of 10 promoter CGIs, and the Spearman correlation is calculated based on original data for each promoter CGI. Comparison of correlation (between the traditional method and CHALM) $P$ values calculated by permutation (Methods section): $<1 \times 10^{-4}$. **b** Similar to **a** but focusing on low-methylation genes. Comparison of correlation permutation $P$ values: $<1 \times 10^{-4}$. **c** Scatter plots show the correlation between H3K4me3 ChIP-seq intensity and methylation level calculated by the traditional and CHALM methods. Balanced promoter CGIs are used. Comparison of correlation permutation $P$ values: $<1 \times 10^{-4}$. **d** Similar to **c** but focusing on low-methylation genes. Comparison of correlation permutation $P$ values: $<1 \times 10^{-4}$. **e**, **f** Methylation status of reads mapped to the promoter CGI of *HIST2H2BF* or *SSTR5*, respectively. Black circles: mCpG; white circles: CpG.

CGIs. With both the traditional and CHALM methods, we consistently observed a negative Spearman correlation between methylation level and H3K4me3 (Fig. 2c and Supplementary Figs. 3b, d and 8). However, when we focused on genes with low methylation levels, only CHALM-determined methylation was significantly anti-correlated with H3K4me3 level (Fig. 2d and Supplementary Fig. 8), suggesting that the CHALM method provides a better representation of the mutually exclusive relationship between DNA methylation and H3K4me3.

To further illustrate that CHALM better explains transcription activity, we next examined in detail two genes with similar methylation levels as determined by the traditional method but with different transcription activities. We found that most reads mapped to the promoter CGI of *HIST2H2BF* were fully unmethylated (low CHALM-determined methylation level), which explained the high transcription activity (Fig. 2e). A large fraction of reads of the repressed gene *SSTR5* had at least one mCpG, which indicated that the high methylation level as determined by the CHALM method was responsible for

transcription repression (Fig. 2f). Collectively, these results demonstrate that the CHALM method provides better prediction of gene expression.

**CHALM performs best in paired-end and high-depth sequencing dataset.** Since the CHALM method quantifies the ratio of methylated reads, its performance depends on the definition of methylated reads, i.e. reads with at least N mCpG sites. We evaluated CHALM based on varying definitions of methylated reads and found that CHALM performed the best when *N* equals to 1 (Supplementary Fig. 9). In addition, we noticed that CHALM requires an average CpG depth of more than 7× in order to achieve the optimal performance (Supplementary Fig. 10). Furthermore, we found that read length would also influence the CHALM performance. For WGBS datasets, the read length is typically ~100 bp, which is too short to cover the promoter CGI region, thus potentially adversely impacting the CHALM performance (Supplementary Fig. 11). We therefore employed an SVD-based imputation method[17–19] to extend the reads. After

method validation, we extended the reads to different lengths and observed that the CHALM performance improved, approaching a plateau at a read length of 300 bp (Supplementary Figs. 12–14). Given these results, CHALM prefers paired-end sequencing data, as the effective read length is twice that of single-end sequencing data (Supplementary Figs. 15 and 16).

**DNA methylation clonal information is crucial for gene expression prediction.** Next, we demonstrate the importance of clonal information for gene expression prediction by DNA methylation with a sophisticated but intuitive deep learning model. In order to maximize the amount of useful information extracted from high-throughput sequencing data, we processed the raw sequencing data into an image-like data structure in which one channel contained methylation information and the other contained read location information (Supplementary Fig. 17). With this data structure, we can leverage more information for gene expression prediction, such as the distance between the read and the transcription start site and the weight of reads with more than one mCpG. These data are then further processed using a convolutional deep neural network for gene expression prediction. As expected, this deep-learning model outperformed a linear model trained using either traditionally determined or CHALM-determined methylation levels (Fig. 3a and Supplementary Fig. 18a). Notably, the deep-learning model (Fig. 3b, c and Supplementary Figs. 18–20) and CHALM (Supplementary Fig. 21) was markedly compromised after we shuffled the mCpG position (assigning mCpG to random reads) while keeping the total number of mCpGs unchanged but entirely disrupting the clonal information. This result demonstrates the crucial role of clonal information in predicting gene expression. We also demonstrated that this deep-learning prediction model outperforms a previously published method[8] in terms of predicting gene expression based on promoter CGI methylation levels (Supplementary Fig. 22). Finally, it is worth noting that the predicted values output by the deep-learning model should not be used as methylation levels, despite that they have a higher correlation with gene expression than the CHALM-determined methylation levels do. The reason is that the deep-learning model is trained to predict gene expression, and thus its output predicted values rely on not only methylation data but also gene expression data; also, the predicted values are derived only for the prediction purpose but ignore other important biological aspects of DNA methylation.

**CHALM identifies more accurate hypermethylated genes during oncogenesis.** To demonstrate the utility of the CHALM method, we compared it to the traditional method for identifying differentially methylated genes with promoter CGIs in paired cancerous and normal lung tissue samples[20] (Supplementary Data 2). The correlation between differential methylation and differential gene expression was significantly greater when the methylation level was calculated using the CHALM method (Fig. 4a). In addition, the CHALM method not only recovered most of the traditional method-identified hypermethylated genes but also identified a subset of genes that are overlooked by the traditional method. Consistent with studies showing that Polycomb-mediated H3K27me3 pre-marks gene promoters for de novo methylation during tumorigenesis[21–23], the hypermethylated genes identified by both methods were highly enriched with H3K27me3 in normal lung tissue. Interestingly, the CHALM-unique hypermethylated genes were more enriched in H3K27me3 than hypermethylated genes uniquely determined by the traditional method, suggesting that CHALM provides more accurate identification of hypermethylated promoter CGIs (Fig. 4b and Supplementary Figs. 23 and 24). Furthermore, with reads extended by imputation, the CHALM method identifies more hypermethylated promoter CGIs that are also enriched with H3K27me3 (Supplementary Fig. 25).

**CHALM identifies de novo DMRs that are more relevant to the studied underlying mechanisms.** We also demonstrated the utility of the CHALM method for calling de novo differentially methylated regions (DMRs)[24]. Lung adenocarcinoma (LUAD) is

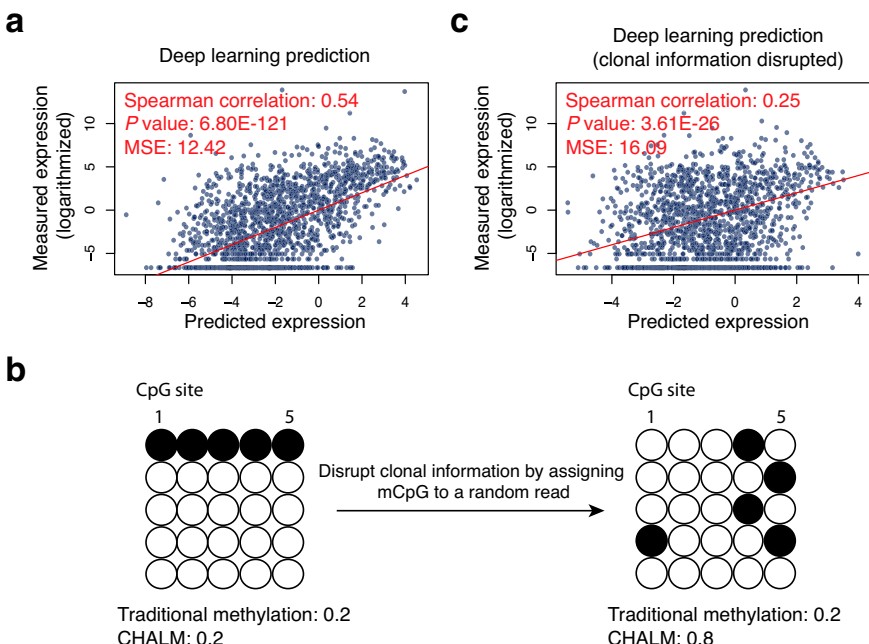

**Fig. 3 Clonal information is crucial for gene expression prediction. a** Prediction of gene expression based on raw bisulfite sequencing reads via a deep-learning framework. **b** Disruption of read clonal information by shuffling the mCpGs among mapped reads. **c** The clonal information is disrupted before prediction. Comparison of correlation (between prediction models with and without clonal information disrupted) permutation $P$ values: $<1 \times 10^{-4}$.

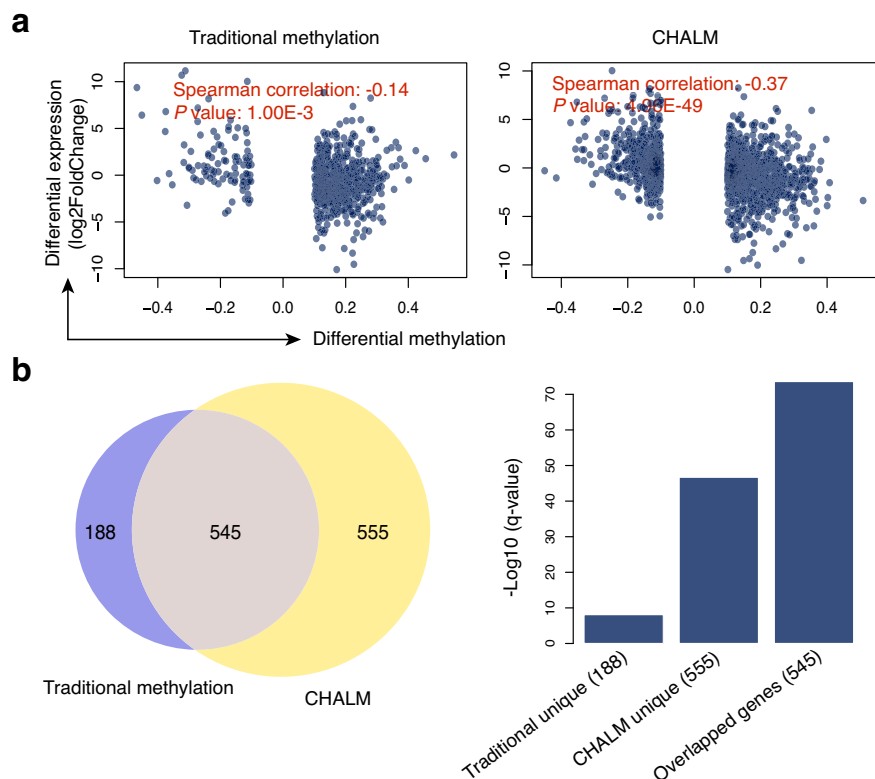

**Fig. 4 CHALM better identifies hypermethylated promoter CGIs during tumorigenesis. a** Scatter plots show the correlation between differential expression and differential methylation calculated by the traditional and CHALM methods. All promoter CGIs were included for analysis, but only those exhibiting a significant methylation change between normal and cancerous lung tissue were plotted. *X*-axis: differential methylation ratio; *y*-axis: differential expression (log2FoldChange). Comparison of correlation (between the traditional method and CHALM) permutation *P* values: <1 × 10⁻⁴. **b** A large fraction of hypermethylated promoter CGIs identified by the traditional method can be recovered using the CHALM method, as indicated by the Venn diagram. Bar plot shows enrichment of the H3K27me3 peak in three different gene sets.

a slow-growing non-small cell lung cancer that accounts for ~40% of lung cancers[25]. Upon treatment with tyrosine kinase inhibitors, a subset of resistant LUADs transform into small cell lung cancer (SCLC), a more-aggressive neuroendocrine tumor[26–28]. To delineate the epigenetic 'rewiring' that underlies this transformation, we called de novo DMRs between LUAD and SCLC using both the traditional and CHALM methods (Supplementary Data 3). We found that a larger fraction of the CHALM unique DMRs were annotated to gene promoter regions (Supplementary Fig. 26). In addition, CHALM-determined hypomethylated DMRs in SCLC were more highly enriched in genes of the neuroactive ligand-receptor interaction pathway, which is reportedly activated in SCLC[29] (Fig. 5a). Expression of genes from this pathway with hypomethylated DMRs was consistently up-regulated in SCLC (Fig. 5b). In addition, a drug repositioning study reported that potential drugs for treating SCLC are enriched in targeting genes associated with the neuroactive ligand-receptor interaction pathway, indicating that this pathway plays a crucial role in SCLC[30]. Collectively, CHALM data suggest that DNA hypomethylation is involved in activating the neuroactive ligand-receptor interaction pathway during the development of SCLC.

Somatostatin receptors (SSTRs) are G-protein-coupled receptors in the neuroactive ligand-receptor interaction pathway that mediate somatostatin's inhibition of cell proliferation, endocrine signaling, and neurotransmission[31]. Given their high expression in neuroendocrine tumors, SSTRs (along with other marker genes) have been used for the detection of neuroendocrine tumors[32,33]. Several somatostatin analogs, including octreotide (SMS 201-995) and vapreotide (RC-160), have been proposed for

use in treating neuroendocrine tumors, including SCLC[34]. Consistent with the up-regulation of SSTR expression in SCLC (Fig. 5c and Supplementary Fig. 27), we also observed significant hypomethylation of the promoter regions of SSTR1, SSTR2, and SSTR5 in SCLC, but only when using the CHALM method (Fig. 5d and Supplementary Fig. 28).

To demonstrate the robustness of the CHALM method, we also used it to identify de novo DMRs during the aging process (mice) and during the development of Alzheimer's disease (humans). In both scenarios, CHALM identified more DMRs closely related to the underlying biological mechanisms (Supplementary Figs. 29 and 30).

## Discussion

We would like to reiterate that CHALM is a method for quantifying cell heterogeneity–adjusted mean methylation, but it is not a method for quantifying methylation heterogeneity per se. Therefore, CHALM is fundamentally different from all of the epigenetic heterogeneity and entropy methods reported before, such as PDR[35], epipolymorphism[36] and Shannon entropy[37]. We compare CHALM and these three heterogeneity methods to the traditional methylation method and note that CHALM exhibited the best correlation with the traditional methylation method. In contrast, the three above-mentioned heterogeneity metrics fit a bell-shaped curve with traditional methylation and thus are not appropriate for direct quantification of methylation, as they cannot distinguish CGIs with low methylation levels (i.e., 0.0–0.2) from those with high methylation levels (i.e., 0.8–1.0; Supplementary Figs. 31–34).

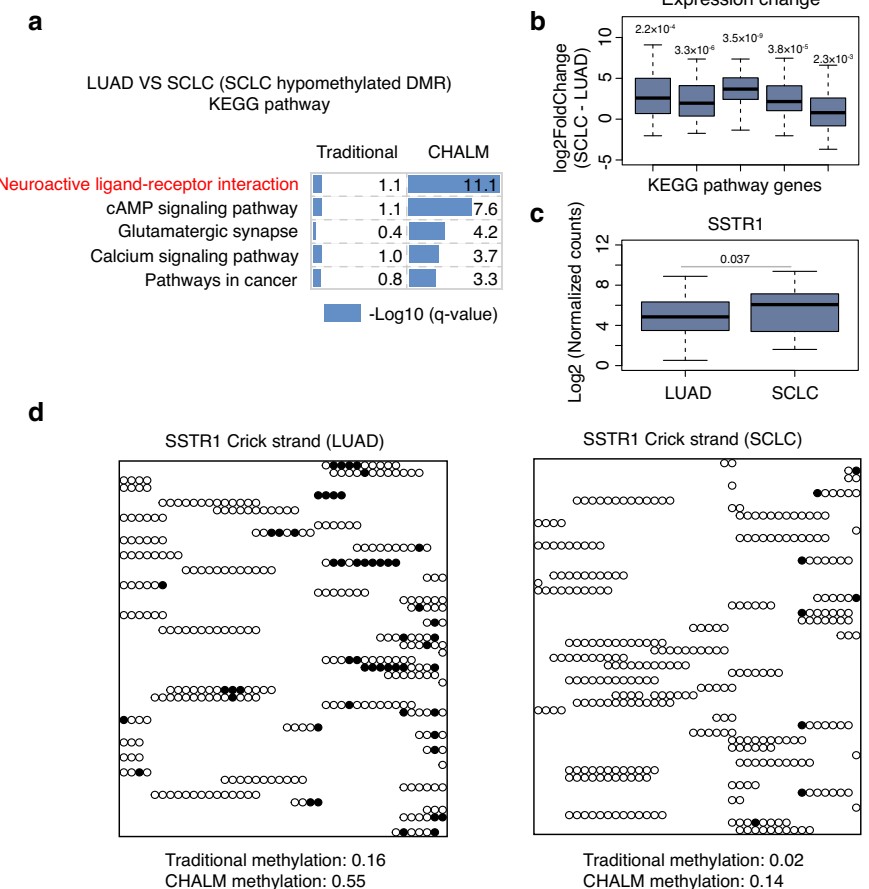

**Fig. 5 CHALM provides better identification of functionally related DMRs. a** KEGG pathway enrichment of the top 2000 hypomethylated DMRs in SCLC. '*q*-value' refers to one-sided Fisher's Exact test *P* value adjusted by Benjamini–Hochberg procedure. **b** Expression change of genes with hypomethylated DMRs in the KEGG pathways shown in **a** between LUAD (79) and SCLC (79) patients. The left-to-right order is the same as the top-to-right order shown in **a**. Two-sided one-sample *t*-test is used. Sample sizes from left to right for test are 57, 41,24, 30, and 49, respectively. **c** Expression of *SSTR1* in LUAD (79) and SCLC (79) patients. Two-sided Wald test *P* value is adjusted by Benjamini–Hochberg procedure. **d** Methylation status of reads mapped to the CHALM-unique hypomethylated DMR found in the *SSTR1* promoter region. Only 50 reads are selected for visualization. The methylation levels shown were calculated based on the original dataset. Black circles: mCpG; white circles: CpG. Boxplot definition: line in the box center refers to the median, the limits of box refer to the 25th and 75th percentiles and whiskers are plotted at the highest and lowest points within the 1.5 times interquartile range.

In conclusion, our data indicate that the CHALM method, which incorporates cell heterogeneity information into DNA methylation quantification, provides a better explanation for the functional consequences of DNA methylation, as evidenced by the demonstrated correlation with gene expression and H3K4me3. We realize that DNA methylation in the promoter region and gene body exhibit different relationships with transcription activity. However, as a causal relationship between gene body methylation and gene expression has not been clearly established[6], we primarily focus on the promoter regions. We further illustrate the importance of clonal information in quantifying DNA methylation using a deep learning model and demonstrate the advantages of the CHALM method for more accurate identification of functionally related DMRs. Finally, although the definition of CHALM involves the ratio of methylated reads, CHALM is actually intended for quantification of the adjusted methylation level for each CpG site, which makes our method compatible with most existing downstream analysis tools, such as differentially methylated cytosine or DMR calling tools (Supplementary Fig. 35). It is anticipated that the CHALM method will be of great value for research that aims to fully delineate the role of DNA methylation in transcription regulation. The CHALM method is available at https://github.com/JianfengXu93/CHALM.

## Methods

**RNA-seq analysis.** Raw sequencing data of CD3 primary cells (GSM1220574), CD14 primary cells (GSM1220575), cancerous and normal lung tissue (GSE70091), and small-cell lung cancer (SCLC, GSE60052) were downloaded from Gene Expression Omnibus (GEO). Raw sequencing data of lung adenocarcinoma (LUAD) samples were downloaded from GDC legacy archive. We used Trimmomatic (0.35)[38] to trim low-quality bases and sequencing adapters. TopHat (2.1.0)[39] was then used to align sequencing reads to the hg19 human reference genome with default parameters. The hg19 GTF annotation file for transcriptome alignment was downloaded from UCSC annotation database. We used Cufflinks (2.2.1)[40] to calculate Fragments Per Kilobase of transcript per Million mapped reads (FPKM) for annotated transcripts. As for differential expression analysis, read counts of transcripts were first calculated by HTSeq (htseq-count, 2.7)[41]. DEseq2 (1.20)[42] was then used to calculate the expression difference and the statistical importance.

**WGBS data pre-processing.** Raw bisulfite sequencing data of CD3 primary cells (GSM1186660), CD4 primary cells (GSM1186661), cancerous and normal lung tissue (GSE70091), and LUAD and SCLC (GSE52271) were downloaded from GEO. After trimming low-quality bases and sequencing adapters, we used BSMAP (2.90)[43] to align reads to hg19 human reference genome with default parameters. The methratio.py (from BSMAP package) script was then used to calculate the methylation ratios of CpG sites. Only CpG sites covered by at least 4 reads are retained for the downstream analyses.

**Promoter CpG islands.** Annotation files for gene position and CpG islands for hg19 assembly were downloaded from UCSC table browser. Promoter CGIs are

defined as CGIs exhibiting overlap with 2-kb windows centered on gene transcription start sites.

**Quantifying the methylation levels of promoter CGIs.** The aforementioned traditional method for calculating promoter methylation level mainly refers to the mean methylation level, which is computed as

$$1/n \sum_{i=1}^{n} C_i/(C_i + T_i) \tag{1}$$

where $C_i$, $T_i$ are the counts of methylated cytosine and unmethylated cytosine on the CpG $i$ of the promoter, respectively.

In our work, we also discussed another traditional method, i.e. weighted methylated level, which is computed as

$$\sum_{i=1}^{n} C_i / \sum_{i=1}^{n} C_i + T_i \tag{2}$$

where $C_i$, $T_i$ are the counts of methylated cytosine and unmethylated cytosine on the CpG $i$ of the promoter, respectively.

The CHALM methylation level is computed as

$$n_{\mathrm{m}}/(n_{\mathrm{m}} + n_{\mathrm{u}}) \tag{3}$$

where $n_{\mathrm{m}}$, $n_{\mathrm{u}}$ are the counts of methylated reads and unmethylated reads mapped to the promoter regions, respectively. Reads with at least one mCpG site are defined as methylated reads.

**Differentially methylated regions (pre-defined regions).** For traditional method, differential methylation of promoter CGIs were calculated by Metilene ('pre-defined regions' mode, 0.2–7) with default parameters.

For CHALM, differential methylation of promoter CGIs were calculated based on beta-binomial model. For a promoter CGI $i$, we denoted the counts of methylated reads, the counts of unmethylated reads and CHALM methylation ratio as $n_{\mathrm{mi}}$, $n_{\mathrm{ui}}$, $p_i$, respectively. The $n_{\mathrm{mi}}$ and $n_{\mathrm{ui}}$ are observed values while $p_i$ is unknown. Given that sequenced reads are sampled from the sequencing cell population, we used binomial distribution to model the methylated reads

$$n_{\mathrm{mi}} \sim \mathrm{B}(n_{\mathrm{mi}} + n_{\mathrm{ui}}, p_i) \tag{4}$$

where the $p_i$ follows a beta distribution $beta(\alpha_i, \beta_i)$, which can be estimated by empirical Bayes method. Similar method has already been implemented in our previously published MOABS package[44]. We then repurposed MOABS to calculate the differential CHALM methylation. The cutoff for significant differential methylation: absolute methylation changes are ≥0.1 and FDR adjusted $p$-value is <0.05.

**Differentially methylated regions (de novo).** For traditional method, de novo DMRs are identified by Metilene ('de novo' mode, 0.2–7) with default parameters.

For CHALM, we first calculated the CHALM methylation ratio for each CpG site. After reads alignment, we scaned each read for mCpG. If a read had at least one mCpG, other CpG sites on the same read would be treated as mCpG as well. Then, the CHALM methylation ratio would be calculated with the methratio.py script from BSMAP. CpG sites covered by at least 4 reads were selected for calling de novo DMRs by Metilene ('de novo' mode).

Identified de novo DMRs by both traditional method and CHALM were annotated to the nearest gene. We then performed pathway enrichment and gene ontology analysis for the differentially methylated genes by using DAVID (6.8) and Enrichr.

**ChIP-seq data analysis.** H3K4me3 ChIP-seq datasets for CD3 primary cell, CD14 primary cell were downloaded from Roadmap project [https://www.ncbi.nlm.nih.gov/geo/roadmap/epigenomics/?view=matrix]. Sequencing reads were aligned to hg19 human reference by bowtie2 (2.2.7, local mode). We then counted mapped reads for each promoter CGI by htseq-count with default setting. Finally, the H3K4me3 ChIP-seq signal intensity of a promoter CGI was defined as read counts normalized by the length of the promoter CGI.

**Balance the promoter CGIs set.** Since most promoter CGIs are unmethylated, the distribution of methylation value of promoter CGIs is severely biased to 0. In order to balance the distribution, all promoter CGIs (~12,000) were split into 200 bins based on their traditional methylation value. For each bin, up to 60 promoter CGIs were randomly selected. The final CGIs set (around 3000 promoter CGIs) is composed of the selected promoter CGIs from 200 bins.

**Permutation test for comparing two correlation coefficients.** Two samples, which have the same size and are used to calculate two Spearman correlation coefficients, $r_1$ and $r_2$, are first pooled into a single sample. In the $b$-th permutation run, we randomly divided this pooled sample into two halves, which would be used to compute two permuted Spearman correlation coefficients, $r_1^{(b)}$ and $r_2^{(b)}$. Then we calculated the difference $r_d^{(b)} = r_2^{(b)} - r_1^{(b)}$. We performed 10,000 independent permutation runs to obtain 10,000 differences under the null hypothesis that the

two samples are from the same distribution: $r_d^{(1)}, \ldots, r_d^{(B)}$. Finally, we compared the original difference $r_d = r_2 - r_1$ to these 10,000 differences to compute a $p$-value defined as $\frac{1}{B}\sum_{b=1}^{B} I(r_d^{(b)} \geq r_d)$ for a one-sided test.

**Missing value imputation.** Since the length of most public bisulfite sequencing datasets is ~100 bp while the length of promoter CGIs ranges from 201 bp to several kb, a single read can only capture a small proportion of CpG sites of a promoter CGI. In order to rescue the information from the uncaptured CpG sites, low-rank SVD approximation (estimated by the EM algorithm) was used to extend the read based on the information of nearby reads[17]. Promoter CGIs larger than 500 bp and with more than 300 mapped reads were selected for imputation. Mapped reads of a promoter CGI were converted into a matrix with column representing CpG sites of this promoter CGI and row representing different reads. Each row contained the methylation status (mCpG: 1; CpG: 0) of CpG sites captured by a single read. The methylation status of the CpG site uncaptured by reads was label as NA and will be imputed by the 'impute.svd' function from bcv package[17,18] (1.0.1).

**Deep learning prediction.** Promoter CGIs with more than 50 mapped reads were selected for deep learning prediction. The methylation status (mCpG: 1; CpG: 0) and the distance of mapped reads to the TSS would be stored into a 3D array. The 3D array is similar to the data structure for storing the positions and pixel information of an image. The first dimension is for storing the mapped reads, which was sorted by the read's methylation fraction

$$f_{\mathrm{m}} = N_{\mathrm{m}}/(N_{\mathrm{m}} + N_{\mathrm{u}})$$

where $N_{\mathrm{m}}$, $N_{\mathrm{u}}$ refers to the number of methylated CpG and unmethylated CpG on this read, respectively. The length of this dimension is 200. When there were less than 200 mapped reads ($N_r < 200$), pseudo-reads were generated by bootstrapping from actual reads. When there were more than 200 mapped reads ($N_r > 200$), $200*F_{\mathrm{size}}$ ($N_r - 200 < 200 \times F_{\mathrm{size}} < N_r$) reads were randomly selected. Selected reads were then sorted based on methylation fraction and split into 200 bins, with $F_{\mathrm{size}}$ reads in each bin. Finally, a pseudo-read was generated based on the mean value of each bin. $N_r$ and $F_{\mathrm{size}}$ refer to the number of mapped reads and the size factor, respectively.

The second dimension is to store the methylation status of the CpG sites on the reads. The dimension length is 10, which stores the methylation status of 10 CpG sites from a sequencing read. When there were <10 CpG sites, the methylation status of a read CpG site was expanded to a pseudo-CpG site. When there were more than 10 CpG sites, the methylation levels of adjacent CpG sites were merged (Supplementary Fig. 36).

The last dimension contains two channels: one channel storing methylation information and the other one storing the distance of mapped reads to TSS.

To train this image-like 3D array ($200 \times 10 \times 2$ data), we built a CNN model with PyTorch (version 1.2). Specifically, the input layer is attached to three sequential Conv2d layers along with RELU activation function. The kernel size of the three Conv2d layers is (5,1), (4,1), and (3,1) respectively. The stride for all Con2d layers is (1,1). Since the second dimension of the input data is small, we did not include pooling layer in our model. The final output layer of this CNN model is a linear regression layer. And in order to prevent overfitting, a dropout layer ($p = 0.2$) was added between the convolution layer and the fully connected layer. We then trained the CNN model using Adam as optimizer and MSELoss as loss function in batches of 32 promoter CGIs.

In order to disrupt the clonal information in the control group, we randomly assigned the mCpGs to mapped reads but kept the total number of mCpGs unchanged. We then sorted the reads based on the methylation fraction to obtain the input matrix, which was used for prediction.

Since most promoter CGIs are unmethylated, the original dataset was downsampled to generate a relatively evenly-distributed dataset (balanced promoter CGI set). Downsampled datasets were then randomly split into training set and test set in a manner of 50–50%. After converting the raw bisulfite sequencing reads into the aforementioned 3D matrix, we trained a convolutional neural network (CNN) model to predict gene expression based on this matrix (Supplementary Fig. 8). The testing set was then used to evaluate the performance of this model.

As a contribution to the community, we also generated a pretrained CNN model by using the RNA-seq and WGBS datasets of 23 different normal tissues from the Roadmap epigenomic project. This pretrained model is ready to use for studying the relationship between DNA methylation and gene expression in other datasets that are of researchers' interest.

**Reporting summary.** Further information on research design is available in the Nature Research Reporting Summary linked to this article.

## Data availability

Public datasets from GEO, TCGA, and Roadmap project are used in this study (Supplementary Data 4). For RNA-seq data, we used CD3 primary cells (GSM1220574), CD14 primary cells (GSM1220575), cancerous and normal lung tissue (GSE70091) and small cell lung cancer (SCLC, GSE60052). The RNA-seq data of lung adenocarcinoma (LUAD) samples were downloaded from GDC legacy archive. For WGBS data, we used CD3 primary cells (GSM1186660), CD4 primary cells (GSM1186661), cancerous and

normal lung tissue (GSE70091), and LUAD and SCLC (GSE52271). H3K4me3 ChIP-seq datasets for CD3 primary cell, CD14 primary cell were downloaded from Roadmap project [https://www.ncbi.nlm.nih.gov/geo/roadmap/epigenomics/?view=matrix].

## Code availability
The source code of the CHALM method is available at https://github.com/JianfengXu93/CHALM. Analysis related to CHALM calculation and gene expression prediction by deep learning is conducted with custom developed tools, which are deposited to the above github website as well. All other custom codes related to this work is available upon request.

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

## Acknowledgements
We acknowledge J. Rosen, H. Liang, L. Shen, N. Weigel, and D. Lamb for constructive discussions and support. This work is funded by the NIH HG007538, NIH R01HL146642, R37CA228304, CA228140, and GM120507.

## Author contributions
J.X., J.P.I., and W.L. conceived and developed the outline of this research. J.X. and J. Shi wrote the tools and performed data analysis and method evaluations. J.X. and X.C. developed the script for deep learning analysis. A.G., X.C., J. Shi, Y.C., and J.J.L. assisted with the manuscript. J.X., J.S., and W.L. wrote the paper.

## Competing interests
After completing the current studies at Baylor College of Medicine, J.X. became a full-time employee at Helio Health. W.L. is a consultant for Helio Health and ChosenMed. The remaining authors declare no competing interests.
