## [Peer Review File · Nature Communications]

Reviewer#4 was asked to respond to points previously raised by Reviewers #1 and #3.

REVIEWER COMMENTS

Reviewer #2 (Remarks to the Author):

The authors responded adequately to my questions and concerns.

Reviewer #4 (Remarks to the Author):

Reviewer 1:

The authors have clearly addressed all questions raised by reviewer 1 except the following two questions:

Reviewer 1 Q2:

This is indeed a very interesting point raised by the previous reviewer and can potentially increase the value of CHALM method. I have several comments regarding the response to this question.

1. The authors claim that the existing single-cell technology is not mature because there is near zero correlation between gene expression and methylation, demonstrated by Figure R1. However, in Youjin Hu, *Genome Biology* 2016, although Figure 3a shows a near-zero correlation between gene expression and methylation in CGI promoters, the correlation is clearly negative in nonCGI promoters shown in Figure 3b. The near-zero correlation observed by the authors is probably due to the much higher proportion of CGI promoters. I would suggest the authors redo the analysis separating CGI and nonCGI, similar to Youjin Hu, *Genome Biology* 2016 Figure 3a, 3b.

2. In the response letter, the authors mentioned the drop-out issue can be a problem leading to the near-zero correlation. I wonder whether imputation methods for scRNA-seq can potentially alleviate this issue?

3. In Figure R1, up to 15 cells are pooled, which may not be enough to create a reliable pseudo-bulk. In GSE76483, there are actually ~50 soma cells and ~100 cytosol cells available. I wonder whether pooling 100 cytosol cells will be able to get a clearer correlation between gene expression and methylation?

4. It seems that the CHALM method improves the performance of various tasks by accounting for cell heterogeneity. However, if there is only one single cell, there should be no cell heterogeneity, and I wonder how the CHALM method compares with other methods. If CHALM still outperforms other methods, what's the potential reason for the improvement given that there is no cell heterogeneity any more?

Reviewer 1 Q4:

1. It still takes some time to understand the paragraph about deep-learning in a bigger picture. The paragraph starts with 'A number of factors must be taken into account when evaluating the power of DNA methylation data for predicting gene expression', which gives the readers an impression that CHALM is going to be used to predict gene expression. However, the purpose of the deep learning model is really to emphasize the importance of clonal information. I would suggest changing the start of the paragraph to something like 'We demonstrate the importance of clonal information by a deep learning model'. Also, the authors do not have a clear definition of what clonal information is throughout the paper (including the name of the CHALM method).

2. I am not quite sure why a deep learning model is necessary to demonstrate the importance of clonal information. Can we simply shuffle the clonal information, recalculate CHALM, and demonstrate that the recalculated CHALM has a lower correlation with the gene expression?

Reviewer 3:

The authors have clearly addressed all questions raised by reviewer 3 and I don't have further comments.

Summary: First, we appreciate that all reviewers found that we had addressed most of their concerns. For example:

- “The authors responded adequately to my questions and concerns.” (Reviewer 2)
- “The authors have clearly addressed all questions raised by reviewer 1 except for the following two questions.” (Reviewer 4)
- “The authors have clearly addressed all questions raised by reviewer 3 and I don’t further comments.” (Reviewer 4)

Second, reviewer 4 appears to have read very carefully our manuscript and our responses to previous reviewers and made multiple **constructive suggestions**. The major recommendations included the following: (1) Clarify the purpose of the deep learning model; (2) Clarify the definition of clonal information; (3) Investigate the CHALM performance after disrupting the clonal information; (4) Use nonCGI promoters and missing value imputation for scRNA-seq analyses.

We believe that we have now satisfactorily addressed all of these concerns in our revised manuscript. In addition, the logic flow of our manuscript has been largely improved thanks to reviewer 4’s insightful suggestions. In summary, we provided the **following one new supplementary figure and three additional reviewer figures**, which we believe significantly strengthen our findings:

1. The performance of CHALM in predicting gene expression after disrupting the clonal information (**Supplementary Fig. 21**).
2. The comparison of CHALM with the traditional method in predicting gene expression from methylation of CGI promoter or nonCGI promoter using a single-cell dataset (**Response Fig. 1**).
3. The performance of CHALM and the traditional method after missing value imputation with ‘sclImpute’ (**Response Fig. 2**).
4. A new figure for pseudo-bulk data merged by 100 single cells (**Response Fig. 3**).

Please find detailed **point-by-point responses** below.

Response to Reviewer #4

Reviewer 1:

The authors have clearly addressed all questions raised by reviewer 1 except the following two questions:

Reviewer 1 Q2:

This is indeed a very interesting point raised by the previous reviewer and can potentially increase the value of CHALM method. I have several comments regarding the response to this question.

1. The authors claim that the existing single-cell technology is not mature because there is near zero correlation between gene expression and methylation, demonstrated by Figure R1. However, in Youjin Hu, Genome Biology 2016, although Figure 3a shows a near-zero correlation between gene expression and methylation in CGI promoters, the correlation is clearly negative in nonCGI promoters shown in Figure 3b. The near-zero correlation observed by the authors is probably due to the much higher proportion of CGI promoters. I would suggest the authors redo the analysis separating CGI and nonCGI, similar to Youjin Hu, Genome Biology 2016 Figure 3a, 3b.

Response: We thank the reviewer for this great suggestion. Consistent with Youjin Hu, Genome Biology 2016 Figure 3a, 3b, we also observed weak negative correlation between methylation and gene expression only for the nonCGI promoters with both traditional and CHALM methylation quantification (**Response Fig. 1**). However, since scRRBS is enriched for high CpG density regions, the read depth in the nonCGI promoter is very low. For a single cell, usually less than 100 genes can pass the filters used by the Youjin Hu Genome Biology 2016. Therefore, the observed negative correlation is not statistically significant with P-values around 0.2. Furthermore, nonCGI promoter has low CpG density with around 1 CpG every 100bp, which is the typical read length from RRBS or WGBS sequencing experiments. In this scenario, CHALM clonal methylation and tradition methylation would be exactly the same with one CpG on the clone (read). This also explains why we cannot find any difference between these two methods on nonCGI promoters.

Response Figure 1. Comparison between the traditional method and CHALM in predicting gene expression with single-cell dataset. Promoters are split into CGI promoters and nonCGI promoters. Results of two representative single cells are shown in **a** and **b**. Pseudo-bulk data with 10 cells or 15 cells merged are shown in **c** and **d**, respectively.

2. In the response letter, the authors mentioned the drop-out issue can be a problem leading to the near-zero correlation. I wonder whether imputation methods for scRNA-seq can potentially alleviate this issue?

Response: We thank the reviewer for this great suggestion. We used the 'scImpute' R package¹ for scRNA-seq imputation. We indeed observed a small improvement of the correlation between DNA methylation and gene expression (**Response Fig. 2**) after missing value imputation, although the improved correlations remain statistically insignificant. Therefore, we cannot draw any meaningful conclusion from this imputation analysis.

Response Figure 2. Comparison between the traditional method and CHALM in predicting gene expression with single-cell dataset (missing value imputed by scImpute). Promoters are split into CGI promoters and nonCGI promoters. Results of two representative single cells are shown in **a** and **b**. Pseudo-bulk data with 10 cells or 15 cells merged are shown in **c** and **d**, respectively.

3. In Figure R1, up to 15 cells are pooled, which may not be enough to create a reliable pseudo-bulk. In GSE76483, there are actually ~50 soma cells and ~100 cytosol cells available. I wonder whether pooling 100 cytosol cells will be able to get a clearer correlation between gene expression and methylation?

Response: We agree with the reviewer that 15 cells may not be enough to create a reliable pseudo-bulk. However, for the GSE76483 dataset, only 15 cells have both scRRBS and scRNA-seq data. To increase the number of cells, we decided to use the GSE121708

dataset from Argelaguet R et al. Nature 2019². After merging 100 cells from GSE121708 to create a pseudo-bulk, we still cannot observe a significant correlation between gene expression and DNA methylation for both the traditional method and CHALM (**Response Fig. 3**). We also noticed that the majority of genes are hypermethylated, which can be partially explained by the global hypermethylation³ in gastrulation cells in GSE121708. Together with Response Figure 1 and 2, we concluded that the single-cell multi-omics technologies are not yet ready for performing the correlation analysis between gene expression and DNA methylation.

Response Figure 3. Comparison between the traditional method and CHALM in predicting gene expression with pseudo-bulk data merged by 100 single cells

4. It seems that the CHALM method improves the performance of various tasks by accounting for cell heterogeneity. However, if there is only one single cell, there should be no cell heterogeneity, and I wonder how the CHALM method compares with other methods. If CHALM still outperforms other methods, what's the potential reason for the improvement given that there is no cell heterogeneity any more?

Response: We thank the reviewer for this great comment. As suggested in Response Figure 1, 2 and 3, the current multi-omics single-cell technologies are not yet ready for performing the correlation analysis between gene expression and DNA methylation. Another reason that CHALM should still outperform the other methods even in single cells might be that CHALM methylation assume that a single mCpG is sufficient for recruiting MBD proteins^{4,5} for gene repression, whereas other methods do not have such strong assumption.

Reviewer 1 Q4:

1. It still takes some time to understand the paragraph about deep-learning in a bigger picture. The paragraph starts with 'A number of factors must be taken into account when evaluating the power of DNA methylation data for predicting gene expression', which gives the readers an impression that CHALM is going to be used to predict gene expression. However, the purpose of the deep learning model is really to emphasize the importance of clonal information. I would suggest changing the start of the paragraph to something like 'We demonstrate the importance of clonal information by a deep learning model'. Also, the authors do not have a

clear definition of what clonal information is throughout the paper (including the name of the CHALM method).

Response: We thank the reviewer for this great suggestion. We reorganized the paragraph as the following: *'Next, we demonstrate the importance of clonal information for gene expression prediction by DNA methylation with a sophisticated but intuitive deep learning model. In order to maximize the amount of useful information extracted from high-throughput sequencing data, we processed the raw sequencing data into an image-like data structure in which one channel contained methylation information and the other contained read location information (Supplementary Fig. 17). With this data structure, we can leverage more information for gene expression prediction, such as the distance between the read and the transcription start site and the weight of reads with more than one mCpG.'*

And we also clarify the definition of 'clonal methylation' in the second paragraph of the main section. *'Clonal methylation here refers to the binary methylation status (methylated or unmethylated) of a genomic locus in a single cell (represented by a read in bisulfite sequencing data).'*

2. I am not quite sure why a deep learning model is necessary to demonstrate the importance of clonal information. Can we simply shuffle the clonal information, recalculate CHALM, and demonstrate that the recalculated CHALM has a lower correlation with the gene expression?

Response: We thank the reviewer for this great suggestion. We shuffled the clonal information and recalculated the CHALM values of the promoter CGIs. The disrupted CHALM values have similar prediction power as the traditional method, suggesting the importance of clonal information for gene prediction (**Supplementary Figure 21**). We still want to keep the deep learning model because it is a more powerful and comprehensive prediction model that considers many different factors. Within such a powerful model, we can still show that clonal information plays such a critical role in gene expression prediction, which strengthens our conclusion.

Supplementary Figure 21. CHALM performance is largely compromised after disrupting the clonal information.

References

- 1 Li, W. V. & Li, J. J. An accurate and robust imputation method scImpute for single-cell RNA-seq data. *Nature communications* **9**, 1-9 (2018).
- 2 Argelaguet, R. *et al.* Multi-omics profiling of mouse gastrulation at single-cell resolution. *Nature* **576**, 487-491 (2019).
- 3 Messerschmidt, D. M., Knowles, B. B. & Solter, D. DNA methylation dynamics during epigenetic reprogramming in the germline and preimplantation embryos. *Genes & development* **28**, 812-828 (2014).
- 4 Ohki, I. *et al.* Solution structure of the methyl-CpG binding domain of human MBD1 in complex with methylated DNA. *Cell* **105**, 487-497 (2001).
- 5 Lewis, J. D. *et al.* Purification, sequence, and cellular localization of a novel chromosomal protein that binds to methylated DNA. *Cell* **69**, 905-914 (1992).

REVIEWERS' COMMENTS:

Reviewer #4 (Remarks to the Author):

The authors have addressed all my comments. I do not have further comments.

REVIEWERS' COMMENTS:

Reviewer #4 (Remarks to the Author): The authors have addressed all my comments. I do not have further comments.

Response: We are grateful for this reviewer's effort in reviewing our manuscript.